# FORGET BUT RECALL: INCREMENTAL LATENT RECTIFICATION IN CONTINUAL LEARNING

## ABSTRACT

Intrinsic capability to continuously learn a changing data stream is a desideratum of deep neural networks (DNNs). However, current DNNs suffer from catastrophic forgetting, which hinders remembering past knowledge. To mitigate this issue, existing Continual Learning (CL) approaches either retain exemplars for replay, regularize learning, or allocate dedicated capacity for new tasks. This paper investigates an unexplored CL direction for incremental learning called Incremental Latent Rectification or ILR. In a nutshell, ILR learns to propagate with correction (or rectify) the representation from the current trained DNN backward to the representation space of the old task, where performing predictive decisions is easier. This rectification process only employs a chain of small representation mapping networks, called rectifier units. Empirical experiments on several continual learning benchmarks, including CIFAR10, CIFAR100, and Tiny ImageNet, demonstrate the effectiveness and potential of this novel CL direction compared to existing representative CL methods.

## 1 INTRODUCTION

Humans exhibit the innate capability to incrementally learn novel concepts while consolidating acquired knowledge into long-term memories (Rasch & Born, 2007). More general Artificial Intelligence systems in real-world applications would require similar imitation to capture the dynamic of the changing data stream. These systems need to acquire knowledge incrementally without retraining, which is computationally expensive and exhibits a large memory footprint (Rebuffi et al., 2016). Nonetheless, existing learning approaches are yet to match human learning in this so-called Continual Learning (CL) problem due to catastrophic forgetting (McCloskey & Cohen, 1989). These systems encounter difficulty balancing the capability of incorporating new task knowledge while maintaining performance on learned tasks, or the plasticity-stability dilemma.

Representative CL approaches in the literature usually involve the use of memory buffer for rehearsal (Ratcliff, 1990; Chaudhry et al., 2019a; Buzzega et al., 2020; Caccia et al., 2022; Bhat et al., 2023; Arani et al., 2022), auxiliary loss term for learning regularization (Kirkpatrick et al., 2017; Ebrahimi et al., 2020; Zenke et al., 2017; Schwarz et al., 2018), or structural changes such as pruning or model growing (Rusu et al., 2016; Mallya & Lazebnik, 2018; Fernando et al., 2017; Yan et al., 2021). These methods share the common objective of discouraging the deviation of learned knowledge representation. Rehearsal-based methods allow the model to revisit past exemplars to reinforce previously learned representations. Alternatively, regularization-based methods prevent changes in parameter spaces by formulating additional loss terms. However, both approaches present shortcomings, including keeping a rehearsal buffer of all past tasks during the model lifetime or infusing ad-hoc inductive bias into the regularization process. Meanwhile, structure-based methods utilize the over-parameterization property of the model by pruning, masking, or adding parameters to reduce new task interferences.

This paper studies a novel approach for CL named Incremental Latent Rectification (ILR), where we allow the model to "forget" knowledge of old tasks but then "recall" or rectify such "catastrophic forgetting" during inference using a sequence of lightweight knowledge mapping networks. These lightweight knowledge mapping networks, called rectifiers, help significantly reduce information loss on learned tasks by incrementally correcting the changes in the representation space. Specifically, for each new task, we add a small, simple, and computationally inexpensive auxiliary unit that will

rectify the representation from the current task to the previous task. Our method differs from many network expansion methods, where additional parameters are allocated to minimize changes to the old parameters. Instead, we iteratively recover past task representations by backwardly propagating current representations through a series of mapping networks. Through this mechanism, ILR allows the optimal adaptation of a new task (plasticity) while separately mitigating catastrophic forgetting. In addition, different from various CL approaches that heavily modify the training process, ILR imposes minimal changes to new task learning as modifications are mainly performed after the training process has been completed. Hence, ILR can be easily integrated into the existing CL pipelines.

**Contributions.** We propose a new direction for CL by sequentially correcting the representation of the current task into the past task's representation using a chain of lightweight rectifier units:

- We propose ILR, a novel approach to continual learning that separates catastrophic forgetting mitigation with new task learning via a sequence of lightweight rectifier units.

- To train the rectifier unit, we rely on either data samples from task $t - 1$ or the current task $t$; when such data is unavailable (e.g., due to memory constraint or privacy concerns), a generative model that synthesizes task $t - 1$'s data can also be utilized. At inference time, for the task-incremental setting, we construct a chain of rectifiers based on the provided task identity and forward the latent representation and inputs to correct the representation. For the class incremental setting, ILR forms the final prediction from an ensemble of predictions based on the reconstructed representations.

- We empirically evaluate our approach on three widely-used continual learning benchmarks (CIFAR10, CIFAR100, and Tiny ImageNet) to demonstrate that our approach achieves comparable performance with the existing representative CL directions.

This paper unfolds as follows. Section 2 discusses the literature on the continual learning problems, and Section 3 describes our Incremental Latent Rectification method. Finally, Section 4 provides the empirical evidence for the effectiveness of our proposed solution.

## 2 RELATED WORK

Catastrophic forgetting is a critical concern in artificial intelligence and is arguably one of the most prominent questions to address for DNNs. This phenomenon presents significant challenges when deploying models in different applications. Continual learning addresses this issue by enabling agents to learn throughout their lifespan. This aspect has gained significant attention recently (Sun et al., 2022; Hu et al., 2021; Kirichenko et al., 2021; Balaji et al., 2020). Considering a model well-trained on past tasks, we risk overwriting its past knowledge by adapting it for new tasks. The problem of knowledge loss can be addressed using different methods, as explored in the literature (Yin et al., 2020; Farajtabar et al., 2020; Kirkpatrick et al., 2017; Li & Hoiem, 2017; Chaudhry et al., 2019a; Bhat et al., 2023; Rusu et al., 2016; Yan et al., 2021) . These methods aim to mitigate knowledge loss and improve task performance through three main approaches: (1) Rehearsal-based methods, which involve reminding the model of past knowledge by using selective exemplars; (2) Regularization-based methods, which penalize changes in past task knowledge through regularization techniques; (3) Parameter-isolation and Dynamic Architecture methods, which allocate sub-networks or expand new sub-networks, respectively, for each task, minimizing task interference and enabling the model to specialize for different tasks.

**Rehearsal-based.** Experience replay methods build and store a memory of the knowledge learned so far (Rebuffi et al., 2016; Lopez-Paz & Ranzato, 2017; Shin et al., 2017; Riemer et al., 2018; Rios & Itti, 2018; Zhang et al., 2019). As an example, Averaged Gradient Episodic Memory (A-GEM) (Chaudhry et al., 2019a) builds an episodic memory of parameter gradients, while ER-Reservoir (Chaudhry et al., 2019) uses a reservoir sampling method to maintain the episodic memory. These methods have shown strong performance in recent studies. However, they require a significant amount of memory to store the examples.

**Regularization-based.** A popular early work using regularization is the elastic weight consolidation (EWC) method (Kirkpatrick et al., 2017). Other methods (Zenke et al., 2017; Aljundi et al., 2018; Van et al., 2022; Nguyen et al., 2018; Ahn et al., 2019) propose different criteria to measure the "importance" of parameters. A later study showed that many regularization-based methods are

variations of Hessian optimization (Yin et al., 2020). These methods typically assume there are multiple optima in the updated loss landscape in the new data distribution. One can find a good optimum for both the new and old data distributions by constraining the deviation from the original model weights.

**Parameter Isolation.** Parameter isolation methods allocate different subsets of the parameters to each task (Rusu et al., 2016; Jerfel et al., 2019; Rao et al., 2019; Li et al., 2019). From the stability-plasticity perspective, these methods implement gating mechanisms that improve stability and control plasticity by activating different gates for each task. Masse et al. (2018) proposes a bio-inspired approach for a context-dependent gating that activates a non-overlapping subset of parameters for any specific task. Supermask in Superposition (Wortsman et al., 2020) is another parameter isolation method that starts with a randomly initialized, fixed base network and, for each task, finds a sub-network (supermask) such that the model achieves good performance.

**Dynamic Architecture.** Different from Parameter Isolation, which allocates subnets for tasks in a fixed main network, this approach dynamically expands the network structure. Yoon et al. (2018) proposes a method that leverages the network structure trained on previous tasks to effectively learn new tasks, while dynamically expanding its capacity by adding or duplicating neurons as needed. Other methods (Xu & Zhu, 2018; Qin et al., 2021) reformulate CL problems into reinforcement learning (RL) problems and leverage RL methods to determine when to expand the architecture when learning new tasks. Yan et al. (2021) introduces a two-stage learning method that first expands the previous frozen task feature representations by a new feature extractor, then re-trains the classifier with current and buffered data.

# 3 PROPOSED FRAMEWORK

We consider the task-incremental and class-incremental learning scenarios, where we sequentially observe a set of tasks $t \in \{1, \ldots, N\}$. The neural network comprises a single task-agnostic feature extractor $f$ and a classifier $w$ with task-specific heads $w^{(t)}|_{t=1}^{N}$. The architecture of $f$ is fixed; however, its parameters are gradually updated as new tasks arrive. At task $t$, the system receives the training dataset $\mathcal{D}_t^{\text{train}}$ sampled from the data distribution $\mathcal{D}_t$ and learns the updated parameters of the feature extractor $f$ and $w$. For easier discussion, the feature extractor and classifier obtained after learning at task $t$ are denoted as $f_t$ and $w_t$, respectively. Thus, after learning on task $t$, we obtain the evolved feature extractor $f_t$ and classifier $w_t$ We call the latent space created by the feature extractor trained with $\mathcal{D}_t^{\text{train}}$ as the $t$-domain. Catastrophic forgetting occurs as the feature extractor $f_{t'}$ is updated into $f_t$, $t' < t$, which causes the $t'$-domain to be overwritten by the $t$-domain. This domain shift degrades the model's performance over time.

To overcome catastrophic forgetting, we propose a new CL paradigm: learning a latent rectification mechanism. This mechanism relies on a lightweight rectifier unit $r_t$ that learns to align the representations from the $t$-domain to the $(t-1)$-domain. Intuitively, this module "corrects" the representation change of a sample from the old task $t-1$ due to the evolution of the feature extractor $f$ when learning the newer task $t$. These rectifier units will establish a chain of corrections for the representation of any task's input, allowing the model to predict the rectified representation better. Figure 1 provides a visualization of the inference process on a task-$t$ sample, after learning $N$ tasks.

Learning the latent rectification mechanism is central to our proposed framework. In general, each rectifier unit should be small compared to the size of the final model or the feature extractor $f$, and its learning process should be resource-efficient. The following sections present and describe our solution for learning this mechanism.

## 3.1 LEARNING THE RECTIFIER UNIT

As the training dataset $\mathcal{D}_t^{\text{train}}$ of task $t$ arrives, we first update the feature extractor $f_t$ and the classifier head $w_t$. The primary goal herein is to find $(f_t, w_t)$ that has high classification performance for task $t$, and the secondary goal is to choose $f_t$ that can reduce the catastrophic forgetting on previous tasks. To combat catastrophic forgetting, we will first discuss the objective function for learning the lightweight rectifier unit $r_t$ and the potential alignment training data (or alignment set) $\mathcal{S}_t$.

Figure 1: At task $t$, the feature extractor $f_t$ and classifier head $w_t$ are optimized on the dataset $D_t^{\text{train}}$. During inference for a test sample from task $t$, we forward the input data $x \in D_t^{\text{test}}$ through the feature extractor and classifier head to obtain the logits. After learning all $N$ tasks, the DNN loses performance on task $t$ due to catastrophic forgetting. Therefore, the latent representation $f_N(x)$ is propagated through a series of rectifiers $r_N, \dots, r_{t+1}$ to perform incremental latent rectification and obtained approximated representations $\hat{f}_{N-1}, \dots, \hat{f}_t$. The logits can be obtained by passing the recovered representation to the respective classifier head.

### 3.1.1 ALIGNMENT LOSS

The goal of $r_t$ is to reduce the discrepancy between task $t$'s representation $f_t(x_i)$ and the previous data representation $f_{t-1}(x_i)$, for $x_i \sim \mathcal{D}_{t-1}$; i.e., $r_t(f_t(x_i), x_i) \approx f_{t-1}(x_i)$. One simple choice is the $l_2$ error between $f_t(x_i)$ and $r_t(f_t(x_i), x_i)$. Let $s$ be a function with parameters $\theta_s$ that encodes inputs $x_i$ into its respective past representation in domain $t-1$. We define the alignment loss as:

$$\mathcal{L}_{\text{align}}(\theta_s; s, \mathcal{S}_t, f_{t-1}) = \mathbb{E}_{x_i \sim \mathcal{S}_t} \left[ \|s(x_i) - f_{t-1}(x_i)\|_2^2 \right]. \tag{1}$$

In practice, we could either store the value of $f_{t-1}(x_i)$ together with $x_i$ in memory or $f_{t-1}$ directly.

### 3.1.2 ALIGNMENT SET

The alignment set $\mathcal{S}_t$ is used as the training data for the rectifier unit $r_t$, enabling the rectifier unit to efficiently learn the mapping from the $t$-domain back to the $t-1$-domain. The design of ILR enables several options for selecting the alignment set, including $\mathcal{D}_{t-1}^{\text{train}}$, $\mathcal{D}_t^{\text{train}}$, or a generative method. Table 1 demonstrates the difference of alignment sets.

**Past task $t-1$ data. (ILR-P)** The simplest choice for the alignment set $\mathcal{S}_t$ is the $\mathcal{D}_{t-1}^{\text{train}}$ (i.e., the training data from the previous task $t-1$), which is sampled directly from the task $t-1$'s distribution. With this option, each element in $\mathcal{S}_t$ is a pair $(x_i, \hat{z}_i)$, where $x_i \in \mathcal{D}_{t-1}^{\text{train}}$ is chosen randomly and $\hat{z}_i = f_{t-1}(x_i)$ is the associated latent representation of $x_i$ under the feature extractor $f_{t-1}$. It is worth noting that this option does *not* keep data samples from all past tasks $t \in \{1, \dots, N\}$ like the rehearsal-based methods (Verwimp et al., 2021).

Table 1: At task $t$, different alignment sets require temporarily storing different components of the training process, which impose different trade-offs in terms of performance, number of parameters, and privacy.

| Variation | $t-1$ samples | $f_{t-1}$ | $G_{t-1}$ |
|---|---|---|---|
| ILR-P ($\mathcal{S}_t \subset \mathcal{D}_{t-1}^{\text{train}}$) | ✓ | - | - |
| ILR-C ($\mathcal{S}_t = \mathcal{D}_t^{\text{train}}$) | - | ✓ | - |
| ILR-G ($\mathcal{S}_t \approx \mathcal{D}_{t-1}$) | - | ✓ | ✓ |

**Current task $t$ data. (ILR-C)** Another potential option for $\mathcal{S}_t$ is task-$t$'s data. If we expect the tasks' data not to be completely unrelated, using data from $\mathcal{D}_t^{\text{train}}$ to train $r_t$ is reasonable. As we show in Section 4, we could achieve comparable performance to strong rehearsal-based methods while remaining *data-free* when setting $\mathcal{S}_t = \mathcal{D}_t^{\text{train}}$. Additionally, for this option, since we do not have access to $t-1$-domain data, we need to keep a copy of $f_{t-1}$ to approximate $\hat{z}_i = f_{t-1}(x_i)$ with $x_i \in \mathcal{D}_t^{\text{train}}$.

**Generated task $t-1$ data. (ILR-G)** Generative methods provide a potential option for creating training data for the rectifier unit $r_t$. Instead of keeping the alignment set $\mathcal{S}_t \subseteq \mathcal{D}_{t-1}^{\text{train}}$, we could train a generative neural network $G_{t-1}$ that learns the task $t-1$ distribution. Unlike generative continual learning methods, $G_{t-1}$ only needs to remember the task $t-1$ distribution instead of all past tasks. Thus, LRB can easily integrate with existing generative methods.

In addition, we could fill $\mathcal{S}_t$ with randomly initialized samples. Nonetheless, our experiments indicate that this approach is ineffective. Therefore, we will focus our discussion on the first three options and leave the exploration for other choices of $\mathcal{S}_t$ for future works.

**Distiction from buffer-based methods.** Rehearsal-based methods retain the data from all past tasks $t \in \{1, \ldots, N\}$ during the lifetime of the DNN. In contrast, depending on the choice of alignment set $\mathcal{S}_t$, ILR can be considered strictly data-free if $\mathcal{S}_t = \mathcal{D}_t^{\text{train}}$ (ILR-C) or if it uses additional generative model (ILR-G). When $\mathcal{S}_t \subseteq \mathcal{D}_{t-1}^{\text{train}}$, ILR-P can still be argued as a data-free method since task $t-1$ data is only retained until the end of task $t$.

## 3.2 INCREMENTAL LATENT ALIGNMENT

The latent alignment mechanism relies on a chain of task-specific rectifier units $(r_t)_{t=2}^N$ that aims to correct the distortion of the representation space as the extractor $f$ learns a new task.

### 3.2.1 LATENT ALIGNMENT

For an input $x$ at task $t-1$, its feature representation under the feature extractor $f_{t-1}$ is $f_{t-1}(x)$. One can heuristically define the $(t-1)$-domain as the representation of the input under the feature extractor $f_{t-1}$. Unfortunately, the $(t-1)$-domain is brittle under extractor update: as the subsequent task $t$ arrives, the feature extractor is updated to $f_t$, and the corresponding feature representation of the same input $x$ will be shifted to $f_t(x)$. Likely, the $t$-domain and the $(t-1)$-domain do not coincide, and $f_t(x) \neq f_{t-1}(x)$.

The feature rectifier unit $r_t$ aims to offset this representation shift. To do this, $r_t$ takes $x$, and its $t$-domain representation $f_t(x)$ as input, and it outputs the rectified representation that satisfies

$$r_t(f_t(x), x) \approx f_{t-1}(x), \tag{2}$$

With this formulation, we can effectively minimize the difference between the rectified representation $r_t(f_t(x), x)$ and the original representation $f_{t-1}(x)$. In practice, we only want to train the rectifier unit $r_t$ and retain the learned feature extractor $f_t$; therefore, let $s(x) = r_t(f_t(x), x)$, we can minimize the difference by using $L_{\text{align}}(\theta_{r_t}; s, \mathcal{S}_t, f_{t-1})$ as in Equation (1).

### 3.2.2 RECTIFIER ARCHITECTURE

The proposed rectifier comprises two trainable components: a *weak feature extractor* $h_t$, and a *gate function* $g_t$. The size of the rectifier units increases linearly with respect to the number of tasks, similar to the classification heads. However, since the rectifier unit is lightweight, this is trivial compared to the size of the full model. Figure 2 visualizes the feature rectifier unit. Alternative designs of the rectifier unit that have been explored are provided in the Appendix.

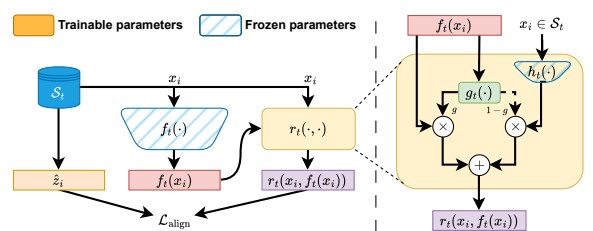

Figure 2: The rectifier unit includes a weak feature extractor $h_t$, and a sigmoid autoencoder $g_t$. The sigmoid autoencoder acts as an element-wise gate function that filters information from $(t-1)$-domain knowledge in $f_t$, while $h_t$ compensates for the loss of information in $f_t$ due to catastrophic forgetting.

**Weak feature extractor $h_t$.** The weak feature extractor $h_t$ processes the input data $x$ to generate a simplified representation $h_t(x)$. $h_t$ is distilled from $f_{t-1}$ to compress the knowledge of $f_{t-1}$ into a more compact, low-capacity parameter-efficient network. For our experiment, we choose the *simplest and most naive* design of a weak feature extractor composed of only two 3x3 convolution layers and two max pooling layers. Instead of processing the full-size image, we use max-pooling to down-sample the input to 16x16 images before feeding into $h_t$. The weak feature extractor is a small network compared to the main model ($h_t$'s architecture is provided in Table 5 in the Appendix).

**Gate function $g_t$.** Due to catastrophic forgetting, the original representation of $f_{t-1}(x)$ will deteriorate as $f$ is updated. The gate function $g_t$ offsets the information loss by computing an element-wise gating weight $0 \leq g_t(f_t(x)) \leq 1$ of the representation $f_t(x)$ to capture only task $t-1$ relevant

information. We use the sigmoid autoencoder similar to TAMiL (Bhat et al., 2023) comprised of a linear encoder with ReLU activation and a linear decoder with sigmoid activation as the gate function.

The weak feature extractor $h_t$ will compensate for the remaining missing information with weight $1 - g_t$. Computing the element-wise weighted average of both representations, we obtain the rectified representation $r_t(x_i, f_t(x_i))$.

$$r_t(x, f_t(x)) = g_t(f_t(x)) \odot f_t(x) + (1 - g_t(f_t(x))) \odot h_t(x) \tag{3}$$

**Distiction from network-expansion approach.** It could be argued that one can, instead, separately train a weak feature extractor $h_t$ for each task, making it a network-expansion CL approach. However, because $h_t$ is a small and low-capacity network, this approach is ineffective; specifically, our experiments demonstrate that the task-incremental average accuracy across all tasks of this approach on CIFAR100 falls below $53\%$. Furthermore, for network expansion approaches, the dedicated parameters are allocated for new task learning, which fundamentally differs from ILR's objective to correct representation changes. The new task's knowledge is acquired by $f_t$ and $w_t$.

## 3.3 TRAINING PROCEDURE

**Network training.** Similar to conventional DNN training, the performance of the feature extractor $f_t$ and the classifier head $w_t$ is measured by the standard multi-class cross-entropy loss:

$$\mathcal{L}_{\text{CE}}(\theta_{f_t}, \theta_{w_t}; f_t, w_t, \mathcal{D}_t^{\text{train}}) = \mathbb{E}_{(x_i, y_i) \sim \mathcal{D}_t^{\text{train}}} \left[ -\sum_{c=1}^{M_t} y_i \log(\hat{y}_i) \right], \tag{4}$$

where $M_t$ is the number of classes of task $t$, $\hat{y}_i$ is the probability-valued network output for the input $x_i$ that depends on the feature extractor $f_t$ and the classifier $w_t$ as $\hat{y}_i = w_t \circ f_t(x_i)$.

Furthermore, we use the past presentation from the alignment set to enforce task $t - 1$ representation consistency, reduce forgetting, and enable more effective rectification by training and regularizing $f_t$ on $\mathcal{D}_t^{\text{train}}$ and $\mathcal{S}_t$, respectively. Let $s(x) = f_t(x)$, then we can similarly use $\mathcal{L}_{\text{align}}(\theta_{f_t}; s, \mathcal{S}_t, f_{t-1})$ in Equation (1) with hyperparameter $\alpha$ :

$$\mathcal{L}_{\text{train}}(\theta_{f_t}, \theta_{w_t}) = \mathcal{L}_{\text{CE}}(\theta_{f_t}, \theta_{w_t}; f_t, w_t, \mathcal{D}_t^{\text{train}}) + \alpha \mathcal{L}_{\text{align}}(\theta_{f_t}; s, \mathcal{S}_t, f_{t-1}). \tag{5}$$

This is different from the rehearsal method since $f$ only visits $\mathcal{D}_{t-1}$ samples at task $t - 1$ and task $t$. After task $t$, $f$ never see $\mathcal{D}_{t-1}$ again, while for rehearsal method, $f$ observe samples from $\mathcal{D}_{t-1}$ throughout its lifetime, risk overfitting on stored exemplars.

**Rectifier training.** Training the rectifier follows two main steps: train the weak feature extractor $h_t$ at task $t - 1$ and then the gate function $g_t$ at task $t$. The weak feature extractor $h_t$ is distilled from $f_{t-1}$ as task $t - 1$ training is completed using $\mathcal{L}_{\text{align}}(\theta_{h_t}; s, \mathcal{D}_{t-1}^{\text{train}}, f_{t-1})$ as in Equation (1) with $s(x) = h_t(x)$. Similarly, after task $t$ training is completed, we also train $g_t$ using $\mathcal{L}_{\text{align}}(\theta_{g_t}; s, \mathcal{S}_t, f_{t-1})$ as in Equation (1) with $s(x) = g_t(f_t(x)) \odot f_t(x)$. Details of ILR's training algorithm are provided in Algorithm 1.

---

**Algorithm 1:** Full training framework at task $t \in \{1, 2, ..., N\}$

---

**Input** : Training dataset $\mathcal{D}_t^{\text{train}}$, hyperparameter $\alpha$, alignment set $\mathcal{S}_t$

1 **for** $\{x_i, y_i\} \in \mathcal{D}_t^{\text{train}}$ **do**
2     Optimize $\theta_{f_t}$ and $\theta_{w_t}$ on $\mathcal{D}_t^{\text{train}}$ with $\mathcal{L}_{\text{train}}(\theta_{f_t}, \theta_{w_t})$ [Equation (5)]
3 **for** $\{x_i, y_i\} \in \mathcal{D}_t^{\text{train}}$ **do**
4     Distill $\theta_{h_{t+1}}$ with $\mathcal{L}_{\text{align}}(\theta_{h_{t+1}}; s, \mathcal{D}_t^{\text{train}}, f_t)$ [Equation (1)] and $s(x) = h_{t+1}(x)$
5 **if** $t > 1$ **then**
6     **for** $\{x_i, f_{t-1}(x_i)\} \in \mathcal{S}_t$ **do**
7         Optimize $\theta g_t$ using $\mathcal{L}_{\text{align}}(\theta_{g_t}; s, \mathcal{S}_t, f_{t-1})$ [Equation (1)] with $s(x) = g_t(f_t(x)) \odot f_t(x)$

---

## 3.4 INFERENCE PROCEDURE

We now describe how to stack multiple rectifier units $r_t$ into a chain for inference. As a new task arrives, our model dynamically extends an additional rectifier unit, forming a sequence of rectifiers.

Table 2: Task-Incremental Average Accuracy across all tasks after CL training. **Joint**: the upper bound accuracy when jointly training on all tasks (i.e., multi-task learning). **Finetuning**: the lower bound accuracy when learning without CL techniques. $|\mathcal{B}|$ is the buffer of all past tasks data, while $|\mathcal{S}_t|$ is the alignment training data set, which only contains data from task $t-1$. NP is the number of parameters (lower is better), and AA is the average accuracy of all tasks (higher is better).

| Method TIL | $|\mathcal{B}|$ | $|\mathcal{S}_t|$ | S-CIFAR10 | | S-CIFAR100 | | S-TinyImg | |
|---|---|---|---|---|---|---|---|---|
| | | | NP | AA | NP | AA | NP | AA |
| Joint | - | - | 11.17M | $98.46_{\pm0.07}$ | 11.22M | $86.37_{\pm0.17}$ | 11.27M | $81.86_{\pm0.57}$ |
| Finetuning | | | 11.17M | $64.16_{\pm2.40}$ | 11.22M | $24.01_{\pm2.14}$ | 11.27M | $13.79_{\pm0.23}$ |
| o-EWC | - | - | 11.17M | $69.60_{\pm5.22}$ | 11.22M | $36.61_{\pm3.82}$ | 11.27M | $15.67_{\pm0.67}$ |
| LwF.mc | | | 11.17M | $60.96_{\pm1.48}$ | 11.22M | $41.00_{\pm1.01}$ | 11.27M | $23.24_{\pm0.71}$ |
| AGEM | 500 | - | 11.17M | $90.37_{\pm1.05}$ | 11.22M | $63.35_{\pm1.47}$ | 11.27M | $37.14_{\pm0.32}$ |
| ER | | | 11.17M | $94.24_{\pm0.24}$ | 11.22M | $67.41_{\pm0.70}$ | 11.27M | $46.07_{\pm0.16}$ |
| DER++ | | | 11.17M | $92.49_{\pm0.55}$ | 11.22M | $68.52_{\pm0.91}$ | 11.27M | $50.84_{\pm0.12}$ |
| ER-ACE | | | 11.17M | $94.52_{\pm0.13}$ | 11.22M | $67.26_{\pm0.50}$ | 11.27M | $47.72_{\pm0.42}$ |
| TAMiL | | | 22.68M | $94.89_{\pm0.16}$ | 22.77M | $76.39_{\pm0.29}$ | 23.20M | $64.24_{\pm0.69}$ |
| CLS-ER | | | 33.52M | $95.35_{\pm0.34}$ | 33.66M | $77.03_{\pm0.81}$ | 33.81M | $54.69_{\pm0.37}$ |
| ILR-P | - | 500 | 12.00M | $86.27_{\pm2.89}$ | 12.05M | $76.23_{\pm0.53}$ | 13.13M | $61.89_{\pm0.15}$ |
| AGEM | 1000 | - | 11.17M | $91.68_{\pm1.48}$ | 11.22M | $67.43_{\pm1.37}$ | 11.27M | $46.94_{\pm0.91}$ |
| ER | | | 11.17M | $95.25_{\pm0.07}$ | 11.22M | $69.69_{\pm1.49}$ | 11.27M | $54.54_{\pm0.40}$ |
| DER++ | | | 11.17M | $93.76_{\pm0.23}$ | 11.22M | $72.27_{\pm1.13}$ | 11.27M | $58.67_{\pm0.28}$ |
| ER-ACE | | | 11.17M | $94.69_{\pm0.25}$ | 11.22M | $72.46_{\pm0.58}$ | 11.27M | $57.37_{\pm0.49}$ |
| TAMiL | | | 22.68M | $95.22_{\pm0.42}$ | 22.77M | $78.72_{\pm0.31}$ | 23.20M | $70.89_{\pm0.04}$ |
| CLS-ER | | | 33.52M | $\mathbf{96.05}_{\pm0.11}$ | 33.66M | $79.36_{\pm0.20}$ | 33.81M | $65.00_{\pm0.02}$ |
| ILR-P | - | 1000 | 12.00M | $90.66_{\pm0.97}$ | 12.05M | $78.14_{\pm0.18}$ | 13.13M | $66.83_{\pm0.55}$ |
| ILR-P | - | 5000 | 12.00M | $92.77_{\pm0.25}$ | 12.05M | $81.50_{\pm0.13}$ | 13.13M | $72.14_{\pm0.43}$ |
| *Alternative alignment sets* | | | | | | | | |
| ILR-C | - | $S_t = D_t$ | - | $89.08_{\pm0.96}$ | - | $79.25_{\pm0.30}$ | - | $66.65_{\pm0.71}$ |
| ILR-G | - | $S_t \sim G_t$ | - | - | - | $81.37_{\pm0.46}$ | - | - |

**Task-Incremental.** We consider a task-incremental learning setting where a test sample $x_i$ is coupled with a task identifier $t_i \in \{1, \dots, N\}$. To classify $x_i$, we can recover $\hat{f}_{t_i}(x)$ by forwarding the current latent variable $f_N(x)$ through a chain of $N - t_i$ rectifiers. We then pass this recovered latent variable through classifier head $w_{t_i}$ to make a prediction. The output $\hat{y}_i$ is computed as

$$\hat{y}_i = w_{t_i}(\hat{f}_{t_i}(x_i)) \quad \text{where} \quad \hat{f}_{t_i}(x_i) = r_{t_i+1}(\hat{f}_{t+i}(x), x) \quad \text{with} \quad t_i < N, \hat{f}_N = f_N$$

**Class-Incremental.** ILR relies on the task identity to reconstruct the appropriate sequence of rectifier units for propagating the latent representation to the original space. However, no identity is provided for the CL method in the class-incremental learning setting. We provided a simple method for inference without task identity, which demonstrates the method's extension to class-incremental learning; however, more robust task-identity inference methods could also be incorporated.

We obtain the class-incremental probabilities by forming an ensemble that averages the class probabilities over all domains. From the current task $t$'s domain, we iteratively rectified the latent back to task $t-1$, task $t-2$, ..., task 1's domain. At each domain, we obtain the rectified representation corresponding with the domain, which we forward through the respective classifier. We then average the softmax probabilities of each domain, essentially forming an ensemble of $w_i(f_i)|_{i=1}^t$.

## 4 EXPERIMENTS

Our implementation [1] is based partially on the Mammoth (Boschini et al., 2022; Buzzega et al., 2020) repository, TAMiL (Bhat et al., 2023) repository, and CLS-ER (Arani et al., 2022) repository.

---

[1]Source code will be publicly released after paper acceptance.

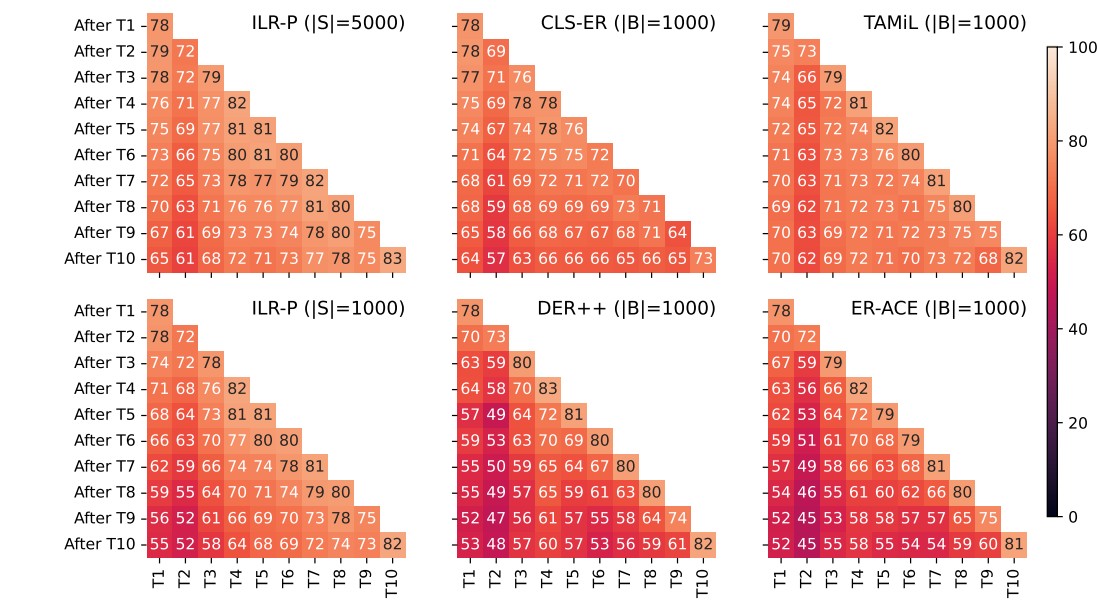

Figure 3: The performance of various CL methods at each task training (lighter color is better). The horizontal axis represents the task on which the model has been trained. The vertical axis represents the task accuracy. ILR-P demonstrates a forgetting rate comparable to or better than other rehearsal-based methods without revisiting past task samples. ILR-P with 1000 sample examples exhibits less forgetting than DER++ and ER-ACE. ILR-P with 5000 sample examples exhibits similar forgetting to CLS-ER and TAMiL.

## 4.1 EVALUATION PROTOCOL

**Datasets.** We select three standard continual learning benchmarks for our experiments: Sequential CIFAR10 (S-CIFAR10), Sequential CIFAR100 (S-CIFAR100), and Sequential Tiny ImageNet (S-TinyImg). Specifically, we divide S-CIFAR10 into five binary classification tasks, S-CIFAR100 into five tasks with 20 classes each, and S-TinyImg into 20 tasks with 20 classes each.

**Baselines.** We evaluate ILR against representative continual learning methods, including EWC (online) (Schwarz et al., 2018), and LwF (multiclass) (Li & Hoiem, 2017), ER (Chaudhry et al., 2019b), AGEM (Chaudhry et al., 2019a), DER++ (Buzzega et al., 2020), ER-ACE (Caccia et al., 2022), CLS-ER (Arani et al., 2022), TAMiL (Bhat et al., 2023). We further provide an upper and lower bound for all methods by joint training on all tasks' data and fine-tuning without catastrophic forgetting mitigation. We employ ResNet18 (He et al., 2016) as the feature extractor for all benchmarks. The classifier comprises a fixed number of separate linear heads for each task.

Table 3: Class-Incremental Average Accuracy across all tasks after CL training. The settings are similar to Table 2.

| Method CIL | $|\mathcal{B}|$ | $|\mathcal{S}_t|$ | S-CIFAR100 NP | AA |
|---|---|---|---|---|
| Joint | - | - | 11.22M | 71.07±0.27 |
| Finetuning | | | 11.22M | 17.50±0.09 |
| DER++ | | | 11.22M | 46.96±0.17 |
| ER-ACE | 1000 | - | 11.22M | 47.09±1.16 |
| TAMiL | | | 22.77M | 51.83±0.41 |
| CLS-ER | | | 33.66M | 51.13±0.12 |
| ILR-P | - | 1000 | 12.25M | 44.45±0.48 |
| ILR-P | - | 5000 | 12.25M | 47.96±0.50 |

Further details on datasets, implementation, and hyperparameters are provided in the Appendix.

## 4.2 RESULTS

**Task-incremental.** Table 2 shows the performance of ILR-P and other CL methods, including rehearsal-based and regularization-based methods, on multiple sequential datasets, including S-CIFAR10, S-CIFAR100, and S-TinyImg. For ILR-P, we create an alignment set from 500, 100, and 5000 samples of $\mathcal{D}_{t-1}^{\mathrm{train}}$. As can be observed from the table, ILR-P achieves comparable results on

S-CIFAR10, compared to the baselines. On S-CIFAR100 and S-TinyImg, ILR-P is equivalent to or outperforms all the baselines, including strong rehearsal-based methods such as TAMiL and CLS-ER, given a sufficient alignment set, indicating its ability to rectify representation changes incrementally.

**Alignment set choices.** Table 2 also demonstrates the results of different alignment set choices. As can be observed, training with data from $\mathcal{D}_{t-1}^{\mathrm{train}}$ (ILR-P) expectedly achieves the best performance since the data is sampled directly from the data distribution $\mathcal{D}_t$ of the previous task; increasing the number of samples from $\mathcal{D}_{t-1}^{\mathrm{train}}$ yields better performance results. The generative network (ILR-G) also yields comparable results due to its ability to synthesize data from $\mathcal{D}_t$. Furthermore, training with $\mathcal{D}_t^{\mathrm{train}}$ (ILR-C) is an attractive choice for its competitive performance and the fact that we do not need to keep a copy of the previous task's data.

**Class-incremental.** Table 3 demonstrates the extension of ILR to class-incremental settings. As the class-incremental probabilities are obtained through averaging, we can still achieve comparable performance to other rehearsal-based methods given a sufficient alignment set.

**Long rectification chain.** Continual learning methods, including rehearsal-based approaches, often experience performance degradation over long task sequences. In Figure 3, we demonstrate that ILR exhibit less forgetting than several continual learning methods across the ten tasks of S-TinyImg.

## 4.3 PARAMETER GROWTH COMPARISON

This section studies the network-size footprint of our framework. The base ResNet-18 has 11.17 million parameters. We report the network sizes after 5, 10, and 20 tasks for ILR and the two baselines, CSL-ER and TAMIL, in Table 4. As we can observe, ILR exhibits a linear memory growth and has the smallest memory footprint among the three baselines. Further analysis reveals that the gate function accounts for 0.06 million parameters while the weak feature extractor account contributes 0.14 million parameters per task.

Table 4: Number of parameters (in millions) of different methods after $N$ tasks measured on the S-TinyImg. The ResNet-18 network with no classifier head is 11.17 million parameters

| Methods | 5 tasks | 10 tasks | 20 tasks |
|---|---|---|---|
| ResNet-18 | 11.27M | 11.27M | 11.27M |
| TAMiL | 22.87M | 23.20M | 23.85M |
| CLS-ER | 33.81M | 33.81M | 33.81M |
| ILR-P | **12.30M** | **13.13M** | **15.41M** |

## 5 LIMITATIONS

We have shown the potential and high utility of ILR's continual learning mechanism in this paper. Nevertheless, ILR also has some limitations. One limitation is that ILR still maintains additional parameters, i.e., the rectifier, which incurs an additional overhead as the number of tasks increases. Inference cost for a long chain would be costly, which can be further explored with modified chaining methods such as skipping (i.e., building a rectifier every two tasks). Additionally, the best performance is achieved with access to task $t-1$'s data. Ideally, we would want to remove this requirement; thus, future research should focus on creating the alignment training data. We have attempted to demonstrate that generative methods are a viable option. Furthermore, since ILR relies on the task identity to reconstruct the rectifier sequence, application to class-incremental learning settings requires either inferring task identity or forming an ensemble of predictions. The proposed ensemble solution might suffer from over-confident or under-confident classifiers. Class-incremental learning is still an open research, where more effective adaptations of our framework can be discovered.

## 6 CONCLUSION

This work proposes a new CL paradigm, ILR, for task incremental learning. ILR tackles catastrophic forgetting through its novel backward-recall mechanism that learns to align the newly learned presentation of past data to their correct representations. Unlike existing CL methods, it requires neither a replay buffer nor intricate training modifications. Our experiments validate that the proposed ILR achieves comparable results to the performance of existing CL baselines for task-incremental and class-incremental learning.

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

## A    DETAILED EXPERIMENTAL SETUP

### A.1    BASELINES

As detailed in Section 4.1, we evaluate ILR against EWC (online version), LwF (multi-class) version, ER, AGEM, DER++, ER-ACE, CLS-ER, and TAMiL.

For extensive comparison, we provide rehearsal-based methods with a buffer with a max capacity of 500 and 1,000 samples, respectively. Since our method does not rely on a buffer of all task data but only an alignment set of task $t - 1$ data, the forgetting can be more significant, which is not a fair comparison of ILR against other rehearsal-based methods. Therefore, we provide ILR with an alignment set of 500, 1,000, and 5,000 samples.

We replicate training settings: For ER, DER++, ER-ACE, TAMiL, and CLS-ER, we employ the reservoir sampling strategy to remove the reliance on task boundaries as in the original implementation. On the other hand, ILR, AGEM, and TAMiL rely on the task boundary to learn the rectifier, modify the buffer, and add a new task-attention module, respectively. For TAMiL, we use the best-reported task-attention architecture. For CLS-ER, we perform inference using the stable model per the original formulation.

### A.2    DATASETS

To demonstrate the effectiveness of our method, we perform empirical evaluations on three standard continual learning benchmarks: Sequential CIFAR10 (S-CIFAR10), Sequential CIFAR100 (S-CIFAR100), and Sequential Tiny ImageNet (S-TinyImg). The datasets are split into 5, 5, and 10 tasks containing 2, 20, and 20 classes, respectively. The dataset of S-CIFAR10 and S-CIFAR100 each includes 60000 $32 \times 32$ images splitter into 50000 training images and 10000 test images, with each task occupying 10000 training images and 2000 testing images. The dataset S-TinyImg contains 1100000 $64 \times 64$ images with 100000 training images and 10000 test images divided into ten tasks with 10000 training images and 1000 test images each. We augment random horizontal flips and random image cropping for each training and buffered image.

### A.3    RECTIFIER DESIGNS

**Weak feature extractor**. We provide the architecture of the weak feature extractor $h_t$ in Table 5. We chose a simple design of two 3x3 convolution layers and two max pooling layers.

Table 5: Architecture of the weak feature extractor $h_t$. We use ReLU activation after each convolution layer. For each task, a weak feature extractor $h_t$ is distilled from the current feature extractor $f_t$.

| Layer | Channel | Kernel | Stride | Padding | Output size |
|-------|---------|--------|--------|---------|-------------|
| Input | 3 | | | | $16 \times 16$ |
| Conv 1 | 64 | $3 \times 3$ | 2 | 1 | $8 \times 8$ |
| MaxPool | | | 2 | | $4 \times 4$ |
| Conv 2 | 128 | $3 \times 3$ | 2 | 1 | $2 \times 2$ |
| MaxPool | | | 2 | | $1 \times 1$ |
| Linear | 512 | | | | |

**Rectifier.** We have explored several options for the rectifier unit design and arrived at a gated rectifier unit, which is the current design, and a compress-combine rectifier unit as in Figure 4. Both designs demonstrate similar performance as shown in Table 6. However, the gated rectifier unit was selected because it is more parameter-efficient than the compress-combine rectifier unit.

The compress-combine design includes a linear layer $a_t$ to reduce $f(t)$'s representation to a lower dimension, essentially forming a bottleneck to filter task $t - 1$ information and a linear layer $b_t$ to combine both $h_t(x)$ representation and reduced $a_t(f_t(x))$ representation.

$$r_t(f_t(x), x) = b_t(\text{concatenate}(a_t(f_t(x)), h_t)) \tag{6}$$

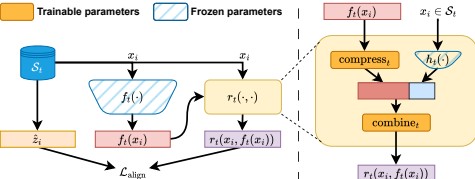

Figure 4: Compress-combine rectifier unit design. The Compress layer forms a bottleneck to select the remaining $(t-1)$-domain knowledge in $f_t$, while $h_t$ extracts compensation information for the loss information in $f_t$. The Combine layer aggregates and transforms the information from both $h_t$ and $f_t$ to form the rectified representation.

Table 6: Average accuracy of ILR-P with $|\mathcal{S}_t| = 1000$ using different rectifier design.

| Rectifier design | S-CIFAR10 | S-CIFAR100 | S-TinyImg |
|---|---|---|---|
| Gated | $90.66_{\pm0.97}$ | $78.14_{\pm0.18}$ | $66.83_{\pm0.55}$ |
| Compress-Combine | $91.02_{\pm1.76}$ | $78.53_{\pm0.25}$ | $66.79_{\pm0.64}$ |

## A.4 TRAINING

**Settings.** The training set of each task is divided into 90%-10% for training and validation. All methods are optimized by the Adam optimizer available in PyTorch with a learning rate of $5 \times 10^{-4}$. As the validation loss plateau for three epochs, we reduce the learning rate by 0.1. Each task is trained for 40 epochs. For ILR, we train $h_t$ and $r_t$ using the same formulation with Adam optimizer at a learning rate of $5 \times 10^{-4}$ for 40 epochs.

**GAN training.** We use the StudioGan repository's default implementation Kang et al. (2023; 2021); Kang & Park (2020) of the BigGAN LeCam Tseng et al. (2021) to train the network on each task of S-CIFAR100. The FID score for each task is between 17 and 23. The BigGAN network has nearly 95 million parameters. During ILR training, we sampled directly from the BigGAN network.

## A.5 HYPERPARAMETER SEARCH

For all methods, experiments, and datasets, we perform a grid search over the following hyperparameters using a validation set. Some hyperparameters are obtained directly from their original implementation to narrow the search range.

- Joint, Finetuning, LwF.mc, ER, AGEM, ER-ACE: No hyperparameters
- o-EWC:
    - $\lambda \in \{10, 20, 50, 100\}$
    - $\gamma \in \{0.9, 1\}$
- DER++:
    - $\alpha \in \{0.1, 0.2, 0.5, 1\}$
    - $\beta \in \{0.1, 0.2, 0.5, 1\}$
- CLS-ER:
    - $r_p \in \{0.5, 0.9\}$
    - $r_s \in \{0.1, 0.5\}$
    - $\alpha_p \in \{0.999\}$
    - $\alpha_s \in \{0.999\}$
- TAMiL:
    - $\alpha \in \{0.2, 0.5, 1\}$
    - $\beta \in \{0.1, 0.2, 1\}$
    - $\theta \in \{0.1\}$
- ILR:
    - $\alpha \in \{1, 2, 3\}$

Table 7: Hyperparameters for method in Table 2

| Method | $|\mathcal{B}|$ | $|\mathcal{S}_t|$ | S-CIFAR10 | S-CIFAR100 | S-TinyImg |
|--------|------|------|-----------|------------|-----------|
| o-EWC | - | - | $\lambda = 100, \gamma = 0.9$ | $\lambda = 50, \gamma = 0.1$ | $\lambda = 20, \gamma = 0.9$ |
| DER++ | | | $\alpha = 0.5, \beta = 0.1$ | $\alpha = 0.2, \beta = 0.1$ | $\alpha = 0.5, \beta = 0.1$ |
| TAMiL | | | $\alpha = 1.0, \beta = 1.0$ | $\alpha = 1.0, \beta = 1.0$ | $\alpha = 1.0, \beta = 0.5$ |
| CLS-ER | | | $r_p = 0.5, r_s = 0.1$ | $r_p = 0.9, r_s = 0.1$ | $r_p = 0.5, r_s = 0.1$ |
| ILR-P | - | 500 | $\alpha = 2$ | $\alpha = 3$ | $\alpha = 3$ |
| DER++ | | | $\alpha = 1.0, \beta = 0.1$ | $\alpha = 0.2, \beta = 0.1$ | $\alpha = 1.0, \beta = 0.1$ |
| TAMiL | | | $\alpha = 1.0, \beta = 1.0$ | $\alpha = 1.0, \beta = 1.0$ | $\alpha = 1.0, \beta = 0.5$ |
| CLS-ER | | | $r_p = 0.5, r_s = 0.1$ | $r_p = 0.5, r_s = 0.1$ | $r_p = 0.9, r_s = 0.1$ |
| ILR-P | - | 1000 | $\alpha = 3$ | $\alpha = 3$ | $\alpha = 3$ |
| ILR-P | - | 5000 | $\alpha = 3$ | $\alpha = 3$ | $\alpha = 3$ |

## B  VERSATILITY OF ILR FRAMEWORK

In ILR, as the tasks arrive, conventional fine-tuning or training on the new task happens without any CL's intervention. ILR only augments or adds to this process with a separate training of the backward-recall mechanism. The attractiveness of this framework is twofold. First, ILR allows the best adaptation on the new task to possibly achieve maximum plasticity while the backward-recall mechanism mitigates catastrophic forgetting. Second, unlike previous CL approaches that heavily modify the sequential training process, ILR minimally changes the fine-tuning process, allowing the users to more flexibly incorporate this framework into their existing machine learning pipelines.

**Relationship to Memory Linking.** ILR's process of mapping newly learned knowledge representation resembles the popular humans' mnemonic memory-linking technique, which establishes associations of fragments of information to enhance memory retention or recall. [2] As the model learns a new task, the feature rectifier unit establishes a mnemonic link from the new representation of the sample from the past task to its past task's correct representation.

---

[2]https://en.wikipedia.org/wiki/Mnemonic_link_system

