# OpenReview forum: "Forget but Recall: Incremental Latent Rectification in Continual Learning"
_ICLR.cc/2025/Conference — ICLR 2025 Conference Withdrawn Submission_

### Official Review · Reviewer_miKU · 2024-10-25

**Soundness:** 3
**Presentation:** 3
**Contribution:** 1
**Rating:** 3
**Confidence:** 4

**Summary:**

This paper proposes Incremental Latent Rectification (ILR) for continual learning. While the main feature extractor weights remain plastic to adapt to new tasks, ILR proposes using a chain of learned feature rectifiers which sequentially map the latent space back to that of previous tasks through a sequence of learned compact feature extractors and gating functions. Because of the compact design of the rectifiers, parameter growth, while linear, is relatively small. To achieve this, these rectifiers must be sequentially aligned with the previous task. Several different alignment set options are proposed: data from the previous task (ILR-P), data from the current task as a surrogate (ILR-C), and synthetic data of the previous task from a learned generator (ILR-G). Experiments are done on sequential versions of CIFAR-10/100 and TinyImageNet, with experiments in both task incremental learning and class incremental learning settings.

**Strengths:**

S1: Novelty: As far as I’m aware, the backward chaining of latent space rectification is relatively novel for continual learning. The rectifier architecture is also somewhat novel, with inspiration that seems to stem from LSTMs or GRUs. At a high level, I can understand where this sort of chaining structure can arise from sequential task learning.

S2. Multiple option for alignment set (Section 3.1.2): The authors propose 3 (actually 4, but random doesn’t work well) ways of creating alignment sets to train each rectifier unit. Given the rectifier design, all three approaches are quite sensible; each have various pros and cons. Depending on the restrictions of a particular deployment setting, one option may be preferable over others, so it’s good to have choices. Table 1 is a good summary of the various differences between the alignment methods, especially their privacy/modeling implications.

S3. Alternative rectifier architectures: There would seem to be many potential rectifier architectures possible, and the overall rectifier chaining strategy doesn’t depend on the specific instantiation (indeed, they can even be different per task, depending on task needs). The Appendix contains an exploration of alternative architectures. On the other hand, I would have liked to see more ablations. For example, is h() even necessary?

S4. Writing: The writing is generally clear in its introduction of the method, and the paper is well-organized, with good visuals. There are a few small errors (see Miscellaneous, under Weaknesses), but nothing that prevents understanding of the paper. One major exception is Section 3.3, where for whatever reason the writing quality drops substantially. I recommend revisiting that section in particular for edits. I’d also suggest taking a look at the notation, which can be at times confusing.

**Weaknesses:**

W1. Architecture design flaws: One of my major concerns is that I’m not convinced of the overall architecture design, for a number of reasons:
- The rectification process becomes linearly slower as the number of tasks increases, due to having to propagate through a chain of rectifiers. For task incremental learning settings, this meaning propagating back to whatever task a sample is coming from; in class incremental learning settings, it’s worse, as not knowing the task ID means having to backpropagate through the entire chain every time, as outline in Section 3.4.
- Due to its chain structure, ILR is highly vulnerable to any single poorly learned rectifier, which would break the chain and make earlier tasks unlearnable. This could practically occur if a particular task has low data, or is particularly out of domain.
- Like many expansion-like CL methods, the proposed ILR becomes less practical for long sequences (experiments consider only up to 20), and likely unusable for continuous stream continual learning settings.
- In addition to all of the above weaknesses, the other major question that leaps out to me is why this sequential rectification process is even necessary. Why not just have a separate independent rectifier per task? That’s essentially what Side-Tuning [a] does. Without the need to chaining, inference time is constant w.r.t. the number of tasks, and any failure for a single task doesn’t doom the prior ones.

W2. Parameter efficiency: While the rectifier compactness is meant to be one of the major selling points of ILR, in practice they’re actually still pretty sizable. On ResNet-18, they represent an almost 10% increase in model parameters after 10 tasks, and as mentioned earlier, will also impact inference time (see Q3). In contrast, see [b] for a method that is able to add feature transformation networks throughout the entire feature extractor (not just the end), but significantly more compactly.

W3. Alignment set design: While I do appreciate the thoroughness of multiple alignment set options, I still see a few weaknesses:
- ILR-P: Saving data from task $t-1$ is still saving data, and not much different from a replay buffer (which typically saves data across all tasks). In the strictest sense, continual learning aims to learn changing data without having to save data from the prior distribution; in many settings, even saving data from the most recent tasks is still not allowable.
- $f_{t-1}$: Both ILR-C and ILR-G use the previous feature extractor for alignment. I see a couple problems with this. a) If $f$ is large, this effectively doubles the model size during training (though not inference). b) If task $t$ is significantly different from $t-1$, there’s a good chance that the data from $t$ will not adequately represent the latent space from $t-1$, limiting the effectiveness of the rectifier alignment.


W4. Experiments: Overall, my general sentiment looking at the main experimental results is that the proposed method’s performance doesn’t distinguish itself compared to the baselines. While not every new method needs to smash the old state-of-the-art, the proposed method doesn’t convincingly outperform the baseline methods (some from 7-8 years ago), even with considerations of model size or buffer size. Some more specific comments:
- Unfair comparison on replay size: I found it a little strange that the size of the alignment set for certain ILR-P results had an alignment set of  $\mathcal S_t = 5000$, but was compared with baselines using a buffer size of 1000. Given the large impact that buffer size can have, this renders comparison with such a row unfair. I suspect this was done because the more equitable comparison at $\mathcal S_t = 1000$ looks less favorable for ILR-P, particularly in the harder class-incremental setting (though that is also the case for S-CIFAR-10). Also disappointing is even with the advantage of a higher buffer size, ILR-P is unable to convincingly claim state of the art results.
- Most experiments are on the task incremental learning setting instead of class incremental learning. Part of this is due to ILR not seeming to be a good fit for CIL, as acknowledged by the authors (Section 5).
- I’m a little disappointed that there is only really one result for ILR-G (task incremental S-CIFAR-100). Given that it’s given a prominent part of the methodology presentation, I was hoping to see more results here.

Miscellaneous:
- Fig 2: It may be helpful to indicate what $s(x)$ is in this figure
- Line 264: “we choose the simplest and most naïve” <= Simpler architectures are possible. Maybe rephrase this as “a simple and naïve”
- Line 278: “Distiction”
- Line 293: “probability-valued network output” => “predicted output probability”
- Line 295: “presentation” <= was this supposed to be “representation”
- Line 302: “risk” => “risking”,
- Line 302: “see” => “sees”
- $s(x)$: I find it confusing that s is being defined to mean different things. There’s probably a better way to introduce this in a technically rigorous way.
- Table 2: Should bold highest value from each AA column
- Line 374: Space before footnote
- Line 482: “it requires neither a replay buffer nor intricate training modifications” <= This claim feels like a stretch. The $t-1$ samples saved by ILR-P is effectively a buffer, with the main distinction that it exclusively comes from the most recent previous task. Similar “intricate” is a vague term. While I agree it’s not that complex, Alg 1 still has more steps than many other continual learning methods.

[a] Zhang et al. “Side-Tuning: A Baseline for Network Adaptation via Additive Side Networks.” ECCV 2020. \
[b] Verma et al. “Efficient feature transformations for discriminative and generative continual learning.” CVPR 2021.

**Questions:**

Q1: How sensitive is ILR-P to the number of samples from $t-1$ saved? A buffer size of 1000 is still pretty big.

Q2: Alg 1: Do you train for a single epoch per task, once for $f_t$ and $w_t$, and once for $h_t$?

Q3: What are the quantitative timings of the proposed method? It would be interesting to see how long inference takes for task incremental learning and class incremental learning settings, as a function of number of tasks.

---

### Official Review · Reviewer_S57f · 2024-11-02

**Soundness:** 3
**Presentation:** 2
**Contribution:** 2
**Rating:** 5
**Confidence:** 4

**Summary:**

The paper presents Incremental Latent Rectification (ILR), a novel approach to continual learning that addresses catastrophic forgetting in deep neural networks. ILR utilizes lightweight rectifier units to correct and align representations from new tasks back to previous tasks, enabling effective knowledge retention. Unlike traditional methods that rely on memory buffers or complex training modifications, ILR integrates seamlessly into existing frameworks. Empirical results on benchmarks like CIFAR10 and Tiny ImageNet demonstrate ILR's competitive performance compared to established continual learning techniques, highlighting its potential for efficient incremental learning without extensive memory requirements.

**Strengths:**

The Incremental Latent Rectification (ILR) approach effectively addresses the issue of catastrophic forgetting in deep neural networks by utilizing lightweight rectifier units. This allows the model to incrementally correct and align representations from new tasks back to previous tasks, ensuring better retention of past knowledge without the need for extensive memory buffers.

ILR integrates seamlessly into existing continual learning pipelines with minimal modifications required during the training process. This flexibility allows practitioners to adopt ILR without overhauling their current systems, making it a practical solution for real-world applications.

Competitive Performance on Benchmarks: Empirical evaluations on standard continual learning benchmarks, such as CIFAR10, CIFAR100, and Tiny ImageNet, demonstrate that ILR achieves performance comparable to or better than established continual learning methods. This indicates its robustness and effectiveness in various learning scenarios.

**Weaknesses:**

1. Ambiguity in Abbreviations (LRB and ILR): The abbreviations LRB and ILR used in the abstract of ICLR web are not clearly defined, which could lead to confusion. Ensuring that these terms are either defined in context or avoided in the abstract would improve clarity and help readers understand the contributions without ambiguity.

2. Limited Evaluation on Diverse Datasets: The experiments primarily focus on three standard benchmarks. While these datasets are widely used, the lack of evaluation on more diverse or complex datasets may limit the generalizability of the Incremental Latent Rectification (ILR) method's effectiveness in real-world applications.

3. Absence of Long-Term Evaluation: The experiments do not address the long-term performance of the ILR method over extended task sequences. Continuous learning scenarios often involve many tasks, and the paper does not provide insights into how well ILR maintains performance over a larger number of tasks or over time, which is critical for assessing its robustness.

4. Reliance on Task Identity: The ILR method's performance is contingent on the availability of task identity during inference, particularly in class-incremental settings. The experiments do not sufficiently explore alternative strategies for inferring task identity or handling situations where task identity is not available, which could limit the method's applicability in practical scenarios.

5. Lack of Open-Source Code: The absence of code or detailed implementation instructions limits the reproducibility and practical application of the ILR method. Providing the code would enhance the transparency of the work, allowing researchers to replicate the results and facilitating further exploration of the method's potential in diverse continual learning scenarios.

**Questions:**

How does the ILR method handle parameter growth when applied to large-scale datasets, such as Full-ImageNet, which involve significantly more classes and complexity? What are the potential limitations or strengths of ILR when applied to datasets with different characteristics (e.g., higher resolution, more complex patterns, or domain-specific features)?

How does the ILR method maintain its performance as the number of tasks increases significantly? What are the scaling limitations of ILR in terms of the maximum number of tasks it can effectively handle?

What is the impact of extended task sequences on memory efficiency and computational overhead?

How can the ILR method be adapted to work effectively in task-agnostic scenarios? What is the performance impact when task identification must be inferred rather than explicitly provided?

---

### Official Review · Reviewer_4a5e · 2024-11-03

**Soundness:** 2
**Presentation:** 3
**Contribution:** 2
**Rating:** 3
**Confidence:** 5

**Summary:**

This paper presents an Incremental Latent Rectification (ILR) method for continual learning, which learns a chain of backward lightweight rectifiers to approximate the latent representation of a previous task using the final feature backbone trained on all tasks. In learning, the backward 2-streaming-task latent alignment loss is based on minimizing the $\ell_2$ error. To train the latent alignment, three choices are used for the alignment sample set (examples sampled from the previous task and their latent representations, examples of the current task and the saved checkpoint of the previous feature backbone, and a generative model trained on the previous task). In experiments, the proposed method is tested under both task-incremental and class-incremental settings, where the class-incremental setting is implemented by averaging predictions from all tasks. It is evaluated on three datasets: CIFAR-10 as 5 binary streaming tasks, CIFAR-100 as 5 20-class tasks, and TinyImageNet as 20 20-class tasks. It obtains reasonably good performance.

**Strengths:**

+ The motivation of exploring incremental latent representation alignment with lightweight implementation is good.
+ The proposed alignment loss and its usage in learning are straightforward.
+ The paper is easy to follow.

**Weaknesses:**

- Overall, the proposed ILR relies on a chain of rectifiers in recovering the latent representation of a previous task from the final feature backbone. There are several drawbacks to be addressed: 1) It's unclear how to prevent error magnification or distributional drift during the chain; 2) The proposed $\ell_2$ error based alignment loss seems less optimal for retaining the classification accuracy on the previous tasks, since the importance of different dimensions in latent features is subject to the classifier weights; 3) The class-incremental setting uses simple averaging strategy which relies on the less practical assumption that all the tasks have the same number of classes in continual learning.
- The proposed method is tested with small deep neural networks (ResNet-18) on streaming tasks with equal number of classes in each task. It also assumes that different tasks are not very different. These limits the applicability and potential significance of the proposed method. For example, it is unclear if the proposed method can handle continual learning on benchmarks with domain shifts such as the VDD benchmark [1].  It is also unclear if the proposed lightweight rectifier will be expressive enough when a larger backbone is used, e.g., Vision Transformer. The rectifiers are learned with significantly downsampled ($16\times 16$) images to be efficient, which seems to be limited in representation learning.  Since a common evolving backbone and lightweight rectifiers are used, there may exist strong limitations, from the perspectives of representational learning and the underlying shear complexity of continual learning in practice,  of the upper bound performance in continual learning, especially for a relative large number classes.
- There are some prior art methods missing in discussions and/or task-incremental comparisons:  Supermasks in Superposition (SupSup) [2], Efficient Feature Transformation (EFT) [3] and Lightweight Learner [4] and learn-to-grow [5]. Comparisons with those method may be helpful in accessing the potential significance of the proposed ILR.

---
[1] Sylvestre-Alvise Rebuffi, Hakan Bilen, and Andrea Vedaldi. Learning multiple visual domains with residual adapters. In Isabelle Guyon, Ulrike von Luxburg, Samy Bengio, Hanna M. Wallach, Rob Fergus, S. V. N. Vishwanathan, and Roman Garnett (eds.), Advances in Neural Information Processing Systems 30: Annual Conference on Neural Information Processing Systems 2017, December 4-9, 2017, Long Beach, CA, USA, pp. 506–516, 2017a. URL https://proceedings.neurips.cc/paper/2017/hash/e7b24b112a44fdd9ee93bdf998c6ca0e-Abstract.html.

[2] Mitchell Wortsman, Vivek Ramanujan, Rosanne Liu, Aniruddha Kembhavi, Mohammad Rastegari, Jason Yosinski, and Ali Farhadi. Supermasks in superposition. In Hugo Larochelle, Marc’Aurelio Ranzato, Raia Hadsell, Maria-Florina Balcan, and Hsuan-Tien Lin (eds.), Advances in Neural Information Processing Systems 33: Annual Conference on Neural Information Processing Systems 2020, NeurIPS 2020, December 6-12, 2020, virtual, 2020. URL https://proceedings.neurips.cc/paper/2020/hash/ad1f8bb9b51f023cdc80cf94bb615aa9-Abstract.html.

[3] Vinay Kumar Verma, Kevin J. Liang, Nikhil Mehta, Piyush Rai, and Lawrence Carin. Efficient feature transformations for discriminative and generative continual learning. In IEEE Conference on Computer Vision and Pattern Recognition, CVPR 2021, virtual, June 19-25, 2021, pp. 13865–13875. Computer Vision Foundation / IEEE, 2021. doi: 10.1109/CVPR46437.2021.01365. URL https://openaccess.thecvf.com/content/CVPR2021/html/Verma_Efficient_Feature_Transformations_for_Discriminative_and_Generative_Continual_Learning_CVPR_2021_paper.html.

[4] Yunhao Ge, Yuecheng Li, Di Wu, Ao Xu, Adam M. Jones, Amanda Sofie Rios, Iordanis Fostiropoulos, shixian wen, Po-Hsuan Huang, Zachary William Murdock, Gozde Sahin, Shuo Ni, Kiran Lekkala, Sumedh Anand Sontakke, and Laurent Itti. Lightweight learner for shared knowledge lifelong learning. Transactions on Machine Learning Research, 2023b. ISSN 2835-8856. URL https://openreview.net/forum?id=Jjl2c8kWUc

[5] Xilai Li, Yingbo Zhou, Tianfu Wu, Richard Socher, and Caiming Xiong. Learn to grow: A continual structure learning framework for overcoming catastrophic forgetting. In Kamalika Chaudhuri and Ruslan Salakhutdinov (eds.), Proceedings of the 36th International Conference on Machine Learning, ICML 2019, 9-15 June 2019, Long Beach, California, USA, volume 97 of Proceedings
of Machine Learning Research, pp. 3925–3934. PMLR, 2019. URL http://proceedings.mlr.press/v97/li19m.html.

**Questions:**

The reviewer would like to check the authors' rebuttal for the comments in the weaknesses.

---

### Official Review · Reviewer_Xmiz · 2024-11-04

**Soundness:** 2
**Presentation:** 3
**Contribution:** 2
**Rating:** 5
**Confidence:** 4

**Summary:**

The paper proposed a research direction as Incremental Latent Rectification (ILR), which consists of learning a rectifier module to match previous data representation from the current learning model. The authors leveraged a weak feature extractor and a gated function to construct the rectifier unit for each task. The rectifiers are trained with feature distillation. During inference, previous features are reconstructed by the rectifiers and forwarded to the classifier. The authors validate the effectiveness of this design through experiments on representative CL benchmarks.

**Strengths:**

1. The paper is well-motivated and well developed

2. The paper is easy-to-follow, well explained

**Weaknesses:**

The paper might be overclaiming the contribution, and the central idea might have already been explored but not cited or mentioned in the paper. Some experimental comparisons might be missing. Please refer to the Questions sections.

**Questions:**

1. Is Incremental Latent Rectification a new CL direction?

At a higher level, learning rectifiers for each task falls into the category of model expansion of CL, as the overall model does expand, despite the objective (lines 54-58). At a lower level, the proposed method in this paper can also be categorized as a distillation method, i.e. regularization method, as the rectifiers are basically trained with feature distillation to preserve previous knowledge. I find it overclaiming to declare ILR as a new research direction.

2. The core idea might have already been explored.

In [a], the authors proposed to learn a predictor network (equivalent to the rectifier) that maps the current state of the representations to their past state to mitigate forgetting. The core idea is very similar to that of this paper with minor differences in the implementation of rectifiers. I acknowledge that the scenario is different (supervised CL and self-supervised CL). However, it is unfortunate that [a] was not mentioned or discussed in the paper. From my understanding, the main difference lies in the use of rectifiers: either we do not keep previous rectifiers but learn to generate features that are quasi-invariant to the state of the model as in [a] or we train and keep all rectifiers and then reconstruct the previous features for inference as in this paper. I am interested to see which strategy is more effective.

3. The comparison might not be sufficient

While the authors claimed several times the difference between the proposed method w.r.t. replay-based methods, the evaluation is mainly with replay-based methods. I find this evaluation contradictory. Also, the compared regularization-based methods (o-EWC and LwF) are too old. Some more recent model expansion or distillation-based methods might be considered, for instance [b].

4. The main table (Table 2) is based on Task-incremental learning, whereas Class-incremental learning evaluation is much less complete.  I am especially interested in ILR-G, as it might be much more challenging to perform generative replay in the CIL setting. And the performance gain of the proposed method in the CIL setting in Table 3 seems to be diminished.

[a]: Self-Supervised Models are Continual Learners, CVPR2022

[b]: Foster: Feature boosting and compression for class-incremental learning, ECCV2022

**Details Of Ethics Concerns:**

looking good

---

### Note · Authors · 2024-11-22

I have read and agree with the venue's withdrawal policy on behalf of myself and my co-authors.